# AUTOSTACKER: AN AUTOMATIC EVOLUTIONARY HIERARCHICAL MACHINE LEARNING SYSTEM

## ABSTRACT

This work provides an automatic machine learning (AutoML) modelling architecture called Autostacker. Autostacker improves the prediction accuracy of machine learning baselines by utilizing an innovative hierarchical stacking architecture and an efficient parameter search algorithm. Neither prior domain knowledge about the data nor feature preprocessing is needed. We significantly reduce the time of AutoML with a naturally inspired algorithm - Parallel Hill Climbing (PHC). By parallelizing PHC, Autostacker can provide candidate pipelines with sufficient prediction accuracy within a short amount of time. These pipelines can be used as is or as a starting point for human experts to build on. By focusing on the modelling process, Autostacker breaks the tradition of following fixed order pipelines by exploring not only single model pipeline but also innovative combinations and structures. As we will show in the experiment section, Autostacker achieves significantly better performance both in terms of test accuracy and time cost comparing with human initial trials and recent popular AutoML system.

## 1 INTRODUCTION

Machine Learning nowadays is the main approach for people to solve prediction problems by utilizing the power of data and algorithms. More and more models have been proposed to solve diverse problems based on the character of these problems. More specifically, different learning targets and collected data correspond to different modelling problems. To solve them, data scientists not only need to know the advantages and disadvantages of various models, they also need to manually tune the hyperparameters within these models. However, understanding thoroughly all of the models and running experiments to tune the hyperparameters involves a lot of effort and cost. Thus, automating the modelling procedure is highly desired both in academic areas and industry.

An AutoML system aims at providing an automatically generated baseline with better performance to support data scientists and experts with specific domain knowledge to solve machine learning problems with less effort. The input to AutoML is a cleanly formatted dataset and the output is one or multiple modelling pipelines which enables the data scientists to begin working from a better starting point. There are some pioneering efforts addressing the challenge of finding appropriate configurations of modelling pipelines and providing some mechanisms to automate this process. However, these works often rely on fixed order machine learning pipelines which are obtained by mimicking the traditional working pipelines of human experts. This initial constraint limits the potential of machine to find better pipelines which may or may not be straightforward, and may or may not have been tried by human experts before.

In this work, we present an architecture called Autostacker which borrows the stacking Wolpert (1992)Breiman (1996) method from ensemble learning, but allows for the discovery of pipelines made up of simply one model or many models combined in an innovative way. All of the automatically generated pipelines from Autostacker will provide a good enough starting point compared with initial trials of human experts. However, there are several challenges to accomplish this:

- The quality of the datasets. Even though we are stepping into a big data era, we have to admit that there are still a lot of problems for which it is hard to collect enough data, especially data with little noise, such as historical events, medical research, natural disasters and so on. We tackle this challenge by always using the raw dataset in all of the stacking layers

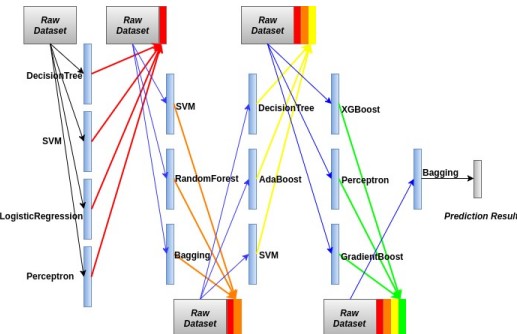

Figure 1: This figure describes the pipeline architecture of Autostacker. Autostacker pipelines consists of one or multiple layers and one or multiple nodes inside each layer. Each node represents a machine learning primitive model, such as SVM, MLP, etc. The number of layers and the number of nodes per layer can be specified beforehand or they can be changeable as part of the hyperparameters. In the first layer, the raw dataset is used as input. Then in the following layers, the prediction results from each node will be added to the raw dataset as synthetic features (new colors). The new dataset generated by each layer will be used as input to the next layer.

while also adding synthetic features in each stacking layer to fully use the information in the current dataset. More details are provided in the Approach section below.

- The generalization ability of the AutoML framework. As mentioned above, existing AutoML frameworks only allow systems to generate an assembly line from data preprocessing and feature engineering to model selection where only a specific single model will be utilized by plugging in a previous model library. In this paper, depending on the computational cost and time cost, we make the number of such primitive models a variable which can be changed dynamically during the pipeline generation process or initialized in the beginning. This means that the simplest pipeline could be a single model, and the most complex pipeline could contain hundreds of primitive models as shown in Figure 1

- The large space of variables. The second challenge mentioned above leads to this problem naturally. Considering the whole AutoML framework, variables include the type of primitive machine learning models, the configuration settings of the framework (for instance, the number of primitive models in each stacking layer) and the hyperparameters in each primitive model. One way to address this issue is to treat this as an optimization problem Feurer et al. (2015). Here in this paper, we instead treat this challenge as a search problem. We propose to use a naturally inspired algorithm, Parallel Hill Climbing (PHC), Ovalle-Martínez et al. (2004) to effectively search for appropriate candidate pipelines.

To make the definition of the problem clear, we will use the terminology listed below throughout this paper:

- Primitive and Pipeline: primitive denotes an existed single machine learning model, for example, a DecisionTree. In addition, these also include traditional ensemble learning models, such as Adaboost and Bagging. The pipeline is the form of the output of Autostacker, which is a single primitive or a combination of primitives.

- Layer and Node: Figure 1 shows the architecture of Autostacker which is formed by multiple stacking layers and multiple nodes in each layers. Each node represents a machine learning primitive model.

## 2 RELATED WORK

Automated Machine Learning has recently gained more attention and there are a variety of related research programs underway and tools coming out. In this section, we first describe recent work in this field and then explain where our work fits in. The current focus in AutoML mainly consists of two parts: machine learning pipeline building and intelligent model hyperparameter search.

Thornton et al. (2013)Kotthoff et al. (2016) provides a software which enables automatic modelling called Auto-Weka. Auto-Weka is built on top of Weka Hall et al. (2009) software and uses Bayesian Optimization (Sequential model-based optimization) to search for optimal hyperparameter settings of the pipeline. The pipeline here follows the traditional machine learning work process: from data preprocessing, feature engineering to single model prediction. However, fixed order pipelines, especially with a single model prediction, are not suitable for complicated problems or small sample datasets. Following the same pipeline, Auto-sklearn Feurer et al. (2015) utilizes the sklearn Pedregosa et al. (2011) machine learning library as a toolbox and searches for the hyperparameter of models with Bayesian Optimization. There are also several works on Bayesian Optimization which are designed specifically for large scale parameter configuration problems like AutoML. For example, RoBO Springenberg et al. (2016) includes multiple implementations of different Bayesian Optimization algorithms with the flexibility of changing the components of this process. HyperoptBergstra et al. (2013) takes advantage of Sequential model-based optimization and considers the choice of classification models and preprocessing models together as an integral optimization problem. Other approaches for large scale parameter search are also included in SMACHutter et al. (2011) and SpearmintSnoek et al. (2012).

By extending the fixed pipeline used in works mentioned above, such as Auto-Weka and Auto-sklearn, one of the most recent and popular framework called TPOT Olson et al. (2016) allows for parallel feature engineering which happens before model prediction. However, all of the works mentioned above follow the same machine learning pipeline. TPOT also uses Evolutionary Algorithms to treat the parameter configuration problem as a search problem. In this work, we use TPOT as one of our baselines.

As described above, there is very little work trying to discover innovative pipelines, even with traditional building blocks, such as sklearn. Pushing the ability of machines to be able to discover innovative machine learning building pipelines, such as new combinations or new arrangements, is necessary to cover a larger space of possible architectures. In this work, we encourage Autostacker to fulfill this requirement in two ways: 1. generating models with new combinations and 2. generating models with completely innovative architectures. In terms of the optimization methods, we offer an alternative solution to search for the settings with Parallel Hill Climbing which is very effective, especially when we are faced with a giant possible search space. The success of using this kind of strategy on large scale AutoML is also proved in TPOT.

## 3 APPROACH

### 3.1 SYSTEM ARCHITECTURE

The working process of Autostacker is shown in Figure 2 and the overview of pipeline architecture built by Autostacker is hown in Figure 1. The whole pipeline consists of multiple layers, where each layer contains multiple nodes. These nodes are the primitive machine learning models. The $i$th layer takes in the dataset $X_i$, and outputs the prediction result $Y_{i,j}$, where $Y_{i,j}$ denotes the prediction result of the $j$th node in the $i$th layer ($i = 0, 1, 2, ..., I$, $j = 0, 1, 2, ..., J$). After each layer's prediction, we add these prediction results back to the dataset used as input to the layer as synthetic features, and then use this newly generated dataset as the input of the next layer. With each new layer, the dataset gets more and more synthetic features until the last layer which only consists of a single node. We take the output of the last layer as the final output of this machine learning problem.

Again, if we use $f_k$ to denote the $k$th ($k = 0, 1, 2, ..., K$) feature in the dataset, the final dataset will contain

$$(K + 1) + \sum_{i=0}^{I-1}(N_i + 1) \tag{1}$$

features in total and this new dataset will be used in the last layer prediction. $N_i$ (0,1,2,...) is the number of nodes in the $i$th layer. The total number of features in the dataset before the last layer can be specified by users.

Unlike the traditional stacking algorithm in ensemble learning, which only feeds the prediction results into next layer as inputs, this proposed architecture always keeps the information directly from the raw dataset. Here are the considerations:

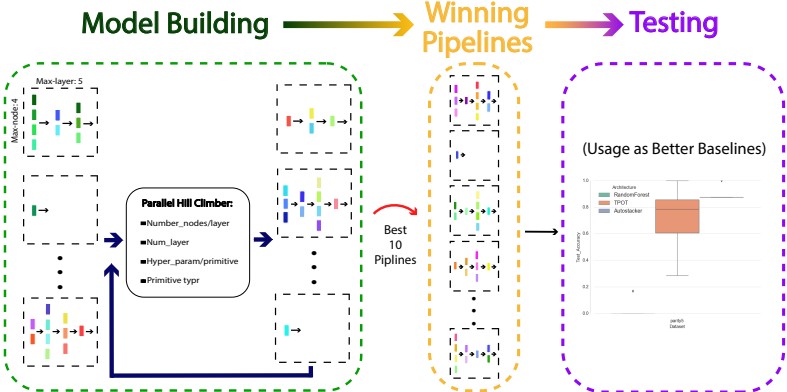

Figure 2: The overview process of Autostacker and its usage is shown here. First, we start to build the model pipelines by generating initial pipelines with dynamic configurations of architecture and hyperparameters, feeding into PHC algorithm, and looping the process to generate winning pipelines. Then the winning pipelines can be used as better baselines for data scientists, or we can analyze the pattern in the winning pipeline sets for further meta-learning usage.

- The number of items in the dataset could be very small. If so, the prediction result from each layer could contain very little information about the problem and it is very likely that the primitives bias the outcomes a lot. Accordingly, throwing away the raw dataset could lead to high-biased prediction results which is not suitable for generalization, especially for situations where we could have more training data in the future.
- Moreover, by combining the new synthetic features with the raw dataset, we implicitly give some features more weight when these features are important for prediction accuracy. Yet we do not delete the raw dataset because we do not fully trust the primitives in individual layers. We can consequently reduce the influences of bias coming from individual primitive and noise coming from the raw dataset.

There are multiple hyperparameters within this architecture:

- $I$ and $J$: the maximum number of layers and the maximum number of nodes corresponding to each layer.
- $H$: the hyperparameters in each primitive.
- The types of the primitives. Here we provide a dictionary of primitives which only serves as a search space.

Note that Autostacker provides two specifications for $I$ and $J$. The default mode is to let users simply specify the maximum range of $I$ and $J$. Only two positive integers are needed to enable Autostacker to explore different configurations. There are two advantages here: 1. This mode frees the system of constraints and allows for the discover of further possible innovative pipelines. 2. This speed up the whole process significantly. We will illustrate this point in the Experiment section later. Another choice is to explicitly denote the value of $I$ and $J$. This allows systems to build pipelines with a specific number of layers and number of nodes per layer based on allowed computational power and time.

The search algorithm for finding the appropriate hyperparameters is described in the next section.

## 3.2 SEARCH ALGORITHM

In this paper, the Parallel Hill Climber (PHC) Algorithm has been chosen as the search algorithm to find the group of hyperparameters which can lead to better baseline model pipelines. PHC is commonly used as baseline algorithm in the development of Evolutionary Algorithm. As we will

show later, our system can already achieve significantly better performance with this straightforward baseline algorithm. Algorithm 1 provides the details of this algorithm in our system.

First, we generate $N$ completed pipelines by randomly selecting the hyperparameters. Then we run a one step Hill Climber on top of these $N$ pipelines to get another $N$ pipelines. The one-step Hill Climber essentially just randomly changes one of the hyperparameters, for example the number of estimators in a Random Forest Classifier. Now we train these $2N$ pipelines and evaluate them through cross validation. Then $N$ pipelines with the highest validation accuracies are selected as the seed pipelines for the next generation of Parallel Hill Climber. Once the seed pipelines are ready, another one step Hill Climber will be applied on them and another round of evaluation and selection will be executed afterwards. The same loop continues until the end of all the iterations, where the number of iterations $M$ can be specified by users.

---

**Algorithm 1** Autostacker Parallel Hill Climber

---

1: $N = 200$
2: $M = 10$
3: $iter\_init = Random(N)$
4: **for** iter in $M$ **do**
5:     $new\_gen = HILLCLIMBER(iter\_init)$
6:     $eva\_pip = iter\_init \cup new\_gen$
7:     $eva\_result = EVALUATE(eva\_pip)$
8:     $sel\_pip = SELECT(eva\_pip, eva\_result, N)$
9:     $iter\_init = sel\_pip$
10: **end for**
11: Return $sel\_pip$
12: **function** HILLCLIMBER($list\_pip$)
13:     **for** each integer $i$ in length of $list\_pip$ **do**
14:         $list\_pip[i] = list\_pip[i]$ with one change
15:     **end for**
16:     Return $list\_pip$
17: **end function**
18: **function** EVALUATE($list\_pip$)
19:     Train the $list\_pip$
20:     **for** each integer $i$ in length of $list\_pip$ **do**
21:         $eva\_result[i] = CV(list\_pip[i])$
22:     **end for**
23:     Return $eva\_result$
24: **end function**
25: **function** SELECT($eva\_pip, eva\_result, N$)
26:     Choose the $Npips$ with highest $eva\_result$
27:     Return $sel\_pip$
28: **end function**

---

### 3.3 TRAINING AND TESTING PROCESS

This section presents the training and testing procedure. The training process happens in the evaluation step as shown above. Corresponding to our hierarchical framework, the pipeline is trained layer by layer. Inside each layer, each primitive is also trained independently with the same dataset. The next layer will be trained after integrating the previous dataset with with prediction results from the previous trained layer. Similarly, the validation process and testing process share the same mechanism but with validation set and test set respectively.

After training and validating the pipelines, we pick the first 10 pipelines with the highest validation accuracies as the final output of Autostacker. We believe that these 10 pipelines can provide better baselines for human experts to get started with the problem. Here choosing the top 10 pipelines instead of the first one pipeline directly is based on the consideration that the input might be a small amount of data which is more likely to be unbalanced. Yet unbalanced data cannot guarantee that the performance on the validation process can fully represent that on the testing process. For example, two pipelines with the same validation results might behave very differently on the same test dataset.

Hence, it is necessary to provide a set of candidates which can be guaranteed to do better on average so that human experts can fine tune the pipelines subsequently.

### 3.4 SCALING AND PARALLELIZATION

Another significant advantage of our approach is that the system is very flexible to scale up and parallelize. Starting from the initial generation, one-step hill climbing, training, validation to evaluation, each pipeline runs independently, which means that each worker can work on one pipeline alone. There is no frequent communication or sequential decision making among all the workers and each worker can run through the pipeline separately. They only need to share the validation result to be ranked by the end of each iteration. Then one shot selection based on the validation accuracy will be applied on the outputs of the parallel workers. More specifically, in terms of the Algorithm 1 we show above, Random(), HILLCLIMBER(), and EVALUATE() function are all very easily parallelized when the system runs.

## 4 EXPERIMENTS

### 4.1 DATASET AND PREPROCESSING

To show the performance of our system, we selected 15 datasets from the benchmark dataset provided in Olson et al. (2017) which collects and cleans datasets from public data resources, such as OpenMLVanschoren et al. (2013) and UCILichman (2013) etc., as the sample experimental data. According to the result published in TPOT, we arbitrarily choose 9 datasets claimed to have better results in TPOT comparing with Random Forest Classifier, 4 datasets with worse performance in TPOT and 2 datasets with same performance with Random Forest Classifier in TPOT. We limit the total number of datasets to be 15 to show here to cover all cases of datasets in TPOT. These datasets come from different problem domains and target different machine learning tasks including binary classification and multi-class classification. Autostacker is also compatible with regression problems. We will release results on more benchmark datasets as well as the code base.

The data is cleaned in terms of filling in the missing values with large negative values or deleting the data points with missing values. Other than that, there is no other data preprocessing or feature preprocessing in Autostacker. It would certainly be possible to use preprocessing on the dataset and features as another building block or hyperparameter in Autostacker, and we also provide this flexibility in our system. Nevertheless, in this paper we focus only on the modelling process to show our contribution to the architecture and automation process. Before each round of the experiment, we shuffle and partition the dataset to $80\%/20\%$ as training/testing data.

### 4.2 BASELINE COMPARISON

The goal of Autostacker is to provide a better baseline pipeline for data scientists in an automatic way. Thus, the baseline we choose to compare with should be able to represent the prediction ability of pipelines coming from the initial trials of data scientists. The baseline pipeline that we compare with is chosen to be Random Forest Classifier / Regressor with the number of estimators being 500 as ensemble learning models like Random Forest have been shown to work well on average in practice when considering multi-model predictions. We also compare our results to those of the TPOT model from Olson et al. (2016) which is one of the most recent and popular AutoML systems.

Currently, our primitives are from the scikit-learn library Pedregosa et al. (2011) and XGboost libaray Chen & Guestrin (2016) The full list is in Table 1 in the appendix. In Autostacker, users are allowed to plug in any primitives they like as long as the function signatures are consistent with our current code base. In terms of the basic structure (number of layers and number of nodes per layer) of the candidate pipelines, as we mentioned above, there are two types of settings provided in Autostacker. In this section, we show the performance of the default mode of Autostacker: dynamic configurations. We specify the maximum range of number of layers to be 5 and the maximum range of number of nodes per layer to be 3.

### 4.2.1 RESULTS

In this section, we will show the results of the test accuracy and time cost of Autostacker as well as comparisons with the Random Forest and TPOT baselines. The test accuracy is calculated using balanced accuracy Velez et al. (2007). We refer to them as test accuracy in the rest of this paper. We ran 10 rounds of experiments for Random Forest Classifier and 3 to 10 rounds of experiment for TPOT based on the computation and time cost. For Autostacker, 3 rounds of experiments are executed on each dataset and the datasets get shuffled before each round. Thus, the figure contains 30 test results in total where each 10 of them come from 1 round experiment. The notches in the box plot represent the 95% confidence intervals of median values. We use one machine with 24 CPUs to parallelize each experiment for each architecture.

As shown in Figure 3, all the left side columns shows the test accuracy comparisons on the 15 sample datasets. From the comparisons, we can tell several things:

- Autostacker achieves **100%** better test accuracy compared with Random Forest Baselines, and 13 out of 15 better accuracy compared with TPOT, while the rest Hill_Valley_with_noise and vehicle datasets achieve extremely similar or slight lower accuracy according to median values.

- Autostacker is much more robust. Autostakcer can always provide better baselines to speed up further work, while Random Forest fails to give any better results on the parity5 dataset, and TPOT fails to provide better baselines than Random Forest Classifier on the breast-cancer, pima, ecoli, wine-recognition and cars datasets after spending a couple of hours searching. This kind of guarantee of not being worse on average comes from the characteristic of Autostacker: the innovative stacking architecture which fully utilizes the predictions coming from different primitive models as well as the whole information of the raw dataset. In the meantime, Autostacker does not give up the single model case if it is better than a stacking architecture. Hence, Autostacker essentially contains the Random Forst Baseline we are using here.

All the right side columns in Figure 3 show the time cost among comparisons. Autostacker largely reduce the time usage up to **6 times** comparing with TPOT on **all the sample datasets**.

In conclusion of the experiment result, the output of Autostacker can improve the baseline pipeline sufficiently enough for human experts to start with better pipelines within a short mount of time, and Autostacker achieves significantly better performance on sample datasets than all baseline comparisons.

## 5 DISCUSSION

During the experiments and research process, we noticed that Autostacker still has several limitations. Here we will describe these limitations and possible future solutions:

- The ability to automate the machine learning process for large scale datasets is limited. Nowadays, there are more sophisticated models or deep learning approaches which achieve very good results on large scale datasets and multi-task problems. Our current primitive library and modelling structure is very limited at solving these problems. One of the future solutions could be to incorporate more advanced primitives and to choose to use them when necessary.

- Autostacker can be made more efficient with better search algorithms. There are a lot of modern evolutionary algorithms, and some of them are based on the Parallel Hill Climber that we use in this work. We believe that Autostacker could be made faster by incorporating them. We also believe traditional methods and knowledge from statistics and probability will be very helpful to better understand the output of Autostacker, such as by answering questions like: why do was a particular pipeline chosen as one of the final candidate pipelines?

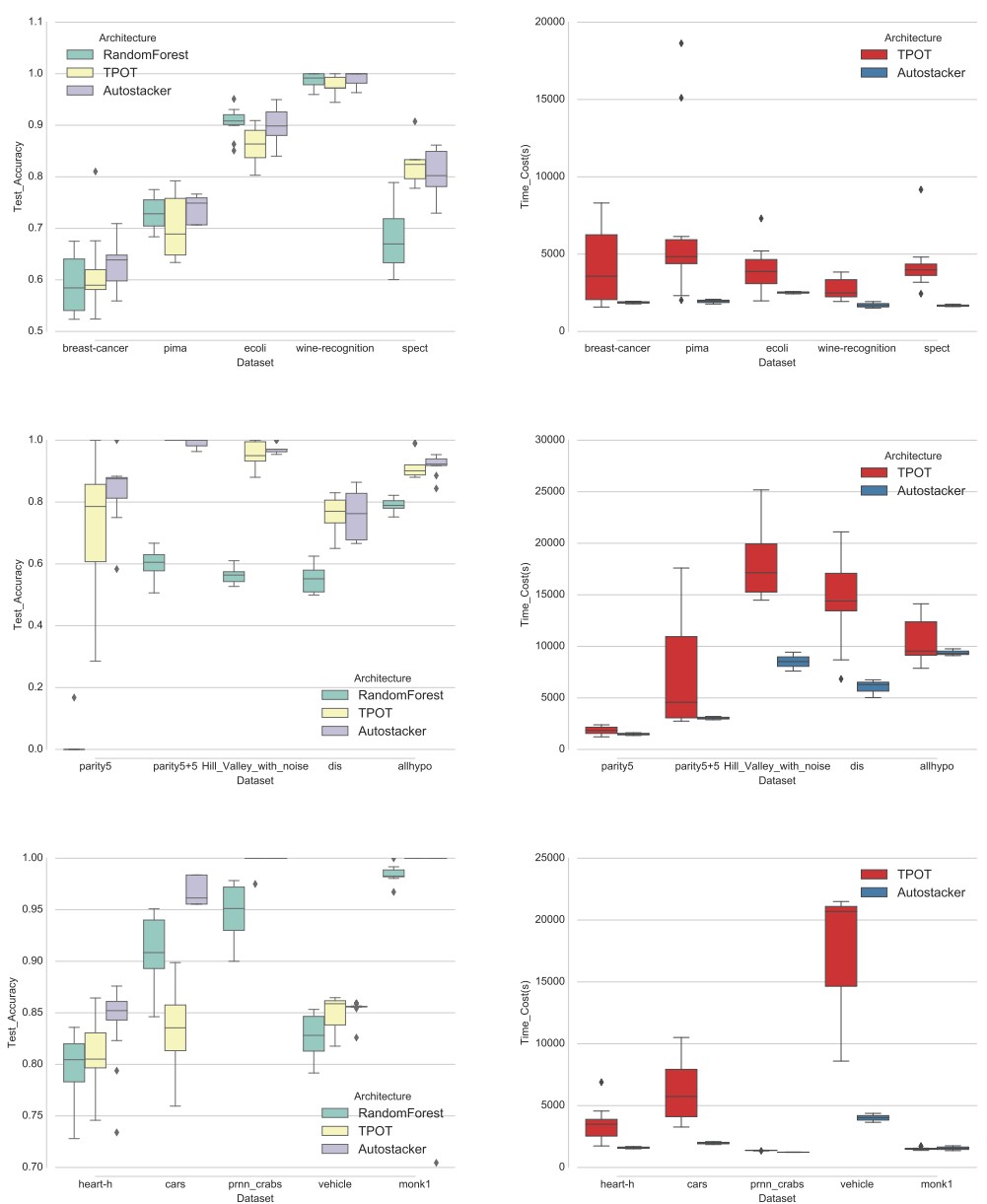

Figure 3: Test Accuracy and Time Cost Comparision.

## 6    CONCLUSION

In this work, we contribute to automating the machine learning modelling process by proposing Autostacker, a machine learning system with an innovative architecture for automatic modelling and a well-behaved efficient search algorithm. We show how this system works and what the performance of this system is, comparing with human initial trails and related state of art techniques. We also demonstrate the scaling and parallelization ability of our system. In conclusion, we automate the machine learning modelling process by providing an efficient, flexible and well-behaved system which provides the potential to be generalized into complicated problems and is able to be integrated with data and feature processing modules.

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

# 7 APPENDIX

Table 1: Primitive List in Autostacker

| Perceptron | AdaBoostClassifier |
|---|---|
| LogisticRegression | XGBClassifier |
| SVC | MLPClassifier |
| DecisionTreeClassifier | BernoulliNB |
| KNeighborsClassifier | MultinomialNB |
| RandomForestClassifier | GradientBoostingClassifier |
| BaggingClassifier | ExtraTreesClassifier |

