# OpenReview forum: "Autostacker: an Automatic Evolutionary Hierarchical  Machine Learning System"
_ICLR.cc/2018/Conference — Reject_

### Official Review · AnonReviewer1 · 2017-11-23
**Not innovative enough, unconvincing results and the writing needs improvement**

**Rating:** 4
**Confidence:** 5

**Review:**

The author present Autostacker, a new algorithm for combining the strength of different learning algorithms during hyper parameter search. During the first step, the hyperparameter search is done in a conventional way. At the second step, the output of each primitives is added to the features of the original dataset and the training and hyperparameter search starts again. This process is repeated for some number of steps. The experiments are performed on 15 small scale dataset and show that Autostacker is performing better  than random forest almost systematically and better than TPOT, the external baseline, 13 times out of 15. Also the speed comparison favor Autostacker vs TPOT.

This algorithm is not highly innovative. Using the output of some algorithms as the input of another one for learning was seen numerous time in the literature. The novelty here is how exactly it is performed, which is a bit ad hoc.

While testing on numerous dataset is important to verify the strength of a learning algorithm, final statistical significance test should be provided e.g. Sign Test, Wilcoxon Signed Rank Test.

The experiment compares with a weak baseline and a baseline that is unknown to me. Also, the datasets are all small scale which is not representative of modern machine learning. This leaves me very uncertain about the actual quality of the proposed algorithm.

The strength of the Random Forest baseline could easily be augmented by simply considering the best learning algorithm over validation across the hyper parameter search (i.e. the choice of the learning algorithm is also a hyperparameter). Also a very simple and fast ensemble could be considered by using Agnostic Bayesian Learning of Ensembles (Lacoste et. al.). It is also common to simply consider a linear combination of the output of the different estimator obtained during cross validation and very simple to implement. This would provide other interesting baselines.

Finally the writing of the paper could be highly improved. Many typos, including several badly formatted citations (consider using \citet and \citep for a proper usage of parenthesis).

---

### Official Review · AnonReviewer2 · 2017-11-27
**Very similar existing work, questionable fit to ICLR**

**Rating:** 3
**Confidence:** 4

**Review:**

In this work, the authors propose to apply parallel hill climbing to learn a stacked machine learning model architecture for a particular machine learning problem. Modest experimental results suggests it compares favorably to another evoluation-inspired AutoML algorithm.

While the idea of Autostacker is presented clearly, the paper has two severe limitations (detailed comments below). First, the contribution itself is fairly minimal; second, even if the contribution were more substantial, the current presentation does not relate to “learning representations” in any meaningful way.

=== Major comments

First, as mentioned above, this paper leans heavily on existing work. Stacking and ensemble methods have become standard approaches in practical machine learning settings (for example, in Kaggle challenges). Likewise, parallel hill climbing (and the closely-related beam search) are common local search strategies for difficult optimization problems. However, it is unclear that combining these yields any unexpected synergies.

Indeed, very similar approaches have been proposed in the literature already. For example, [Welchowski, T. & Schmidt, M. A framework for parameter estimation and model selection in kernel deep stacking networks. Artificial Intelligence in Medicine, 2016, 70, 31-40], propose a very similar model, including search with hill climbing and using the original data at each layer. While they do restrict the considered primitive model type, neither paper offers any compelling theoretical results, so this is largely an implementation detail in terms of novelty.

Additionally, the paper lacks any discussion about how the architectures may change during search, as well as what sorts of architectures are learned. For example, the given number of layers and nodes are maximums; however, the text just above Algorithm 1 points out that the first step in the algorithm is to “generate N completed pipelines.” What exactly does this mean? If PHC randomly changes one of the architecture hyperparameters, what happens? e.g., which layer is removed? Ultimately, what types of architectures are selected?

Finally, ICLR does not seem like a good venue for this work. As presented, the work does not discuss learning representations in any way; likewise, none of the primitive models in Table 1 are typically considered “representation learning models.” Thus, it is not obvious that Autostacker would be especially effective at optimizing the hyperparameters of those models. Experimental results including these types of models could, in principle, demonstrate that Autostacker is applicable, but the current work does not show this.

=== Minor comments

Section 3.3 seems to advocate training on the testing data. Even if the described approach is common practice (e.g., looking at 10-fold CV results, updating the model, and running CV again), selecting among the models using inner- and outer-validation sets would avoid explicitly using information about the testing set for improving the model.

How sensitive is the approach to the choice of the number of layers and nodes, both in terms of accuracy and resource usage?

It would be helpful to include basic characteristics of the datasets used in this study, perhaps as a table in the appendix.

=== Typos, etc.

The paper has numerous typos and needs thorough editing. The references in the text are not formatted correctly. I do not believe this affects understanding the paper, but it definitely disrupts reading.

The references are inconsistently and incorrectly formatted (e.g., “Bayesian” should be capitalized).

---

### Official Review · AnonReviewer3 · 2017-11-27
**Reinvents cascading, optimizes it with hill climbing**

**Rating:** 4
**Confidence:** 5

**Review:**

The authors introduce a simple hill climbing approach to (very roughly) search in the space of cascades of classifiers.
They first reinvent the concept of cascades of classifiers as an extension of stacking (https://en.wikipedia.org/wiki/Cascading_classifiers). Cascading is like stacking but carries over all original model inputs to the next classifier.
The authors cast this nicely into a network view with nodes that are classifiers and layers that use the outputs from previous layers. However, other than relating this line of work to the ICLR community, this interpretation of cascading is not put to any use.
The paper incorrectly claims that existing AutoML frameworks only allow using a specific single model. In fact, Auto-sklearn (Feurer et al, 2015) automatically constructs ensembles of up to 50 models, helping it to achieve more robust performance.

I have some questions about the hillclimbing approach:
- How is the "one change" implemented in the hill climber? Does this evaluate results for each of several single changes and pick the best one? Or does it simply change one classifier and continue? Or does it evaluate all possible individual changes and pick the best one? I note that the term "HillClimber" would suggest that some sort of improvement has to be made in each step, but the algorithm description does not show any evaluation step at this point. The hill climbing described in the text seems to make sense, but the pseudocode appears broken.

Section 4.2: I am surprised that there is only a comparison to TPOT, not one to Auto-sklearn. Especially since Auto-sklearn constructs ensembles posthoc this would be an interesting comparison.
As the maximum range of number of layers is 5, I assume that scaling is actually an issue in practice after all, and the use of hundreds of primitive models alluded to in the introduction are not a reality at this point.

The paper mentions guarantees twice:
- "This kind of guarantee of not being worse on average comes from the the characteristic of AutoStacked"
- "can be guaranteed to do better on average"
I am confident that this is a mistake / an error in choosing the right expression in English. I cannot see why there should be a guarantee of any sort.

Empirically, Autostacker appears better than RandomForest, but that is not a big feat. The improvements vs. TPOT are more relevant. One question: the data sets used in Olson et al are very small. Does TPOT overfit on these? Since AutoStacker does not search as exhaustively, could this explain part of the performance difference? How many models are evaluated in total by each of the methods?

I am unsure about the domain for the HillClimber. Does it also a search over which classifiers are used where in the pipeline, or only about their hyperparameters?

Minor issues:
- The authors systematically use citations wrongly, apparently never using citep but only having inline citations.
- Some parts of the paper feel unscientific, such as using phrases like "giant possible search space".
- There are also several English grammar mistakes (e.g., see the paragraph containing "for the discover") and typos.
- Why exactly would a small amount of data be more likely to be unbalanced?
- The data "cleaning" method of throwing out data with missing values is very unclean. I hope this has only been applied to the training set and that no test set data points have been dropped?
- Line 27 of Algorithm 1: sel_pip has not been defined here

Overall, this is an interesting line of work, but it does not seem quite ready for publication.

Pros:
- AutoML is a topic of high importance to both academia and industry
- Good empirical results

Cons:
- Cascading is not new
- Unclear algorithm: what exactly does the Hillclimber function do?
- Missing baseline comparison to Auto-sklearn
- Incorrect statements about guarantees

---

### Decision · Program_Chairs · 2018-01-29
**ICLR 2018 Conference Acceptance Decision**

**Decision:**

Reject

**Comment:**

The reviewers have pointed out that there is a substantial amount of related work that this paper should be acknowledging and building on.